# Gene Therapeutic Drug pCMV-VEGF165 Plasmid (‘Neovasculgen’) Promotes Gingiva Soft Tissue Augmentation in Rabbits

**DOI:** 10.3390/ijms251810013

**Published:** 2024-09-17

**Authors:** Polina Koteneva, Nastasia Kosheleva, Alexey Fayzullin, Yana Khristidis, Timur Rasulov, Aida Kulova, Sergey Rozhkov, Anna Vedyaeva, Tatiana Brailovskaya, Peter Timashev

**Affiliations:** 1Institute for Regenerative Medicine, Sechenov University, 119991 Moscow, Russia; 2Central Research Institute of Dentistry and Maxillofacial Surgery, 119991 Moscow, Russia; 3Digital Dentistry Center, 129090 Moscow, Russia; 4E.V. Borovsky Institute of Dentistry, Sechenov University, 119991 Moscow, Russia; 5World-Class Research Center “Digital Biodesign and Personalized Healthcare”, Sechenov University, 119991 Moscow, Russia

**Keywords:** angiogenesis, gene therapy, pCMV-VEGF165 plasmid, ‘Neovasculgen’, porcine skin matrix, xenogeneic materials, Mucoderm, gingiva soft tissue augmentation, dentistry, regenerative medicine, tissue engineering, vascularization

## Abstract

Currently, an increasing number of patients are undergoing extensive surgeries to restore the mucosa of the gums in the area of recessions. The use of a connective tissue graft from the palate is the gold standard of such surgical treatment, but complications, especially in cases of extensive defects, have led to the development of approaches using xenogeneic collagen matrices and methods to stimulate their regenerative and vasculogenic potential. This study investigated the potential of a xenogeneic scaffold derived from porcine skin Mucoderm and injections of the pCMV-VEGF165 plasmid (‘Neovasculgen’) to enhance soft gingival tissue volume and vascularization in an experimental model in rabbits. In vitro studies demonstrated the biocompatibility of the matrix and plasmid with gingival mesenchymal stem cells, showing no toxic effects and supporting cell viability and metabolic activity. In the in vivo experiment, the combination of Mucoderm and the pCMV-VEGF165 plasmid (0.12 mg) synergistically promoted tissue proliferation and vascularization. The thickness of soft tissues at the implantation site significantly increased with the combined application (3257.8 ± 1093.5 µm). Meanwhile, in the control group, the thickness of the submucosa was 341.8 ± 65.6 µm, and after the implantation of only Mucoderm, the thickness of the submucosa was 2041.6 ± 496.8 µm. Furthermore, when using a combination of Mucoderm and the pCMV-VEGF165 plasmid, the density and diameter of blood vessels were notably augmented, with a mean value of 226.7 ± 45.9 per 1 mm^2^ of tissue, while in the control group, it was only 68.3 ± 17.2 per 1 mm^2^ of tissue. With the application of only Mucoderm, it was 131.7 ± 37.1 per 1 mm^2^ of tissue, and with only the pCMV-VEGF165 plasmid, it was 145 ± 37.82 per 1 mm^2^ of the sample. Thus, the use of the pCMV-VEGF165 plasmid (‘Neovasculgen’) in combination with the xenogeneic collagen matrix Mucoderm potentiated the pro-proliferative effect of the membrane and the pro-vascularization effect of the plasmid. These results indicate the promising potential of this innovative approach for clinical applications in regenerative medicine and dentistry.

## 1. Introduction

Currently, the deficiency of bone and soft tissues of the gingiva is a serious problem in dentistry, especially in aesthetically significant areas. In patients with a thin gingiva phenotype, the risk of developing single or multiple gingiva recessions after orthodontic treatment remains high. In recent years, classical methods of surgical elimination of gingiva recessions have been described in detail and are constantly being improved and updated. The classical surgical methods of treating recessions used today have their positive and negative sides. The use of a connective tissue graft from the palate is the gold standard of surgical treatment due to excellent clinical results and predictability. However, this method has a disadvantage associated with the creation of an additional wound surface in the donor area of the hard palate, leading to the risk of increased postoperative soreness [1,2,3,4].

Also, to close multiple gingiva recessions, the resource of the donor area of the palate may be limited. A considerable number of complications, such as the risk of damage to a branch of the palatine artery, distant bleeding from the donor area, difficult healing, persistent pain syndrome with damage to the periosteum, and necrosis and paresthesia of the palate, create the need to find an alternative to connective tissue grafts. In order to avoid such complications, numerous approaches are being developed and used, including various hydrogels [5,6], degradable biomedical elastomers [7], and alternative xenogeneic collagen materials [8,9]. Matrices for soft-tissue augmentation must meet the following criteria—have volumetric stability over time, be biocompatible, and have a minimum biodegradation period, which allows for achieving the remodeling process [10].

A significant challenge in the use of artificial xenogeneic matrices is the inadequate vascularization of the newly formed tissue. When these matrices are implanted, the body quickly reacts to the foreign object, but achieving proper vascularization remains difficult due to the complex nature of the process and its low efficiency. The lack of vascularization can result in ischemia and necrosis in the tissue. To address this issue, therapeutic agents such as plasmids with VEGF (vascular endothelial growth factor) gene vectors have emerged as a potential solution. These plasmids have shown promise in improving vascularization in contexts of various diseases [11,12,13,14]. Despite these advancements, it is important to note that the efficacy of *VEGF* gene vectors for dental applications and soft tissue augmentation has not been thoroughly investigated. This gap in the research presents an opportunity for further exploration into the potential benefits and limitations of using *VEGF* gene vectors in these specific contexts. By studying the application of these vectors in dental and soft tissue procedures, researchers can gain a better understanding of their effectiveness and potential impact on clinical outcomes. This could ultimately lead to improvements in the treatment of conditions requiring tissue regeneration and augmentation in the oral and maxillofacial regions.

Based on the conclusions of the experimental study, it is necessary to understand the scope of clinical application of certain membranes as well as the degree of immune response to them in the form of a local inflammatory reaction. All of the above require clinical studies in order to optimize existing surgical methods for the treatment of multiple gingiva recessions, reduce the risk of postoperative complications, reduce the time of surgical intervention, and minimize patient discomfort [15,16,17,18,19,20].

This investigation presents an examination of the biological compatibility and therapeutic potential of the collagen matrix Mucoderm in conjunction with the gene therapy agent pCMV-VEGF165 plasmid (‘Neovasculgen’) through a series of in vitro and in vivo experiments. The study encompasses an assessment of the biocompatibility of the matrix and plasmid on gingival mesenchymal stem cells (MSCs) via viability and metabolic activity assays as well as an evaluation of the capacity of this composite to augment the volume of soft gingival tissues in a rabbit model.

## 2. Results

### 2.1. Biocompatibility of Mucoderm and pCMV-VEGF165 Plasmid

No cytotoxic effects were observed for samples using Mucoderm and the pCMV-VEGF165 plasmid. Analysis of extracts revealed that for all concentrations, there was no decrease in metabolic activity, although mesenchymal stem cell (MSC) proliferation and DNA amount were reduced to approximately 694.3 ± 24.1 and 554.7 ± 21.4 ng/mL in cases of the Mucoderm and ‘Neovasculgen’ extracts (Figure 1A). Conversely, the metabolic activity of cells when cultured in a series of dilutions of the pCMV-VEGF165 plasmid (‘Neovasculgen’) did not significantly decrease at the maximum values of the drug concentration (0.12 mg/mL), up to 98% of the control values (Figure 1A). However, the level of MSC proliferation in the presence of the pCMV-VEGF165 plasmid also decreased two-fold relative to the control samples in the growth medium. In the positive control, exposure of MSC cultures to the Sodium dodecyl sulphate (SDS) solution resulted in a significant decrease in viability and DNA amount, with a lethal concentration of SDS observed at 12.5% dilution.

The obtained results align with the data from contact cytotoxicity analyses of Mucoderm and the pCMV-VEGF165 plasmid. Notably, MSCs cultured for 3 days on the surface did not form a viable monolayer but remained alive on the surface of Mucoderm (Figure 1B). After 7 days of culture on the Mucoderm surface, the cells formed a monolayer (Figure 1C). Furthermore, pure Mucoderm material exhibited a fibrillar structure, and MSCs cultured on Mucoderm’s surface were well spread and displayed typical morphology (Figure 1D,E).

### 2.2. In Vivo Studies of Mucoderm Material and pCMV-VEGF165 Plasmid (‘Neovasculgen’) Vascularization Properties 

When examining samples from the control group, fragments of gingival tissues were lined with stratified squamous epithelium with pronounced papillae (Figure 2a,b). In the submucosa, singular lymphocytes were detected; collagen fibers had a loose structure and were stained red by Masson; a small number of fibroblasts were observed (Figure 3a). The thickness of the submucosa was 341.8 ± 65.6 µm. Blood vessels were rare, and some of them were full-blooded; their density was 68.3 ± 17.2 per 1 mm^2^ of the tissue (Figure 3b). The underlying tissues consisted of adipose tissue with thin fibrous layers containing large dilated full-blooded vessels of various types without perivascular infiltration around, as well as thickened nerves. At the periphery of histological preparations were skeletal muscles and mucoprotein glands.

pCMV-VEGF165 plasmid (‘Neovasculgen’). When using the pCMV-VEGF165 plasmid, an area of proliferating connective tissue was formed de novo at the injection site. Proliferation of the basal layer of the stratified squamous keratinizing epithelium with the formation of branched structures was noted (Figure 2c,d). A connective tissue grew around the foci on the injected pCMV-VEGF165 plasmid, richly nourished by multiple blood vessels of various calibers (Figure 3c). The blood vessels included densely located capillaries and large arteries and veins oriented along the fibers of the connective tissue (Figure 3d). Their density was 145 ± 37.82 per 1 mm^2^ of the sample. The thickness of the submucosa at the site of the pCMV-VEGF165 plasmid (‘Neovasculgen’) injection was significantly increased due to the growth of the connective tissue capsule and amounted to 1366.5 ± 420.6 µm. The increase in tissue was localized, and an increase in the number of collagen fibers and the thickness of their bundles was observed only within the boundaries of the de novo connective tissue, but the average number of blood vessels was increased throughout the entire sample due to capillaries and small-caliber vessels.

In the group of Mucoderm, a pronounced proliferative reaction was observed at the implantation site (Figure 2e,f). The area of membrane implantation was increased somewhat due to the fibers of the matrix itself, which were swelled and multidirectional, but largely due to the formation of dense fibrotic granulation tissue consisting of thick layers of parallel bundles of collagen fibers with a high density of fibroblasts between them (Figure 3e). The thickness of the submucosa was 2041.6 ± 496.8 µm. The tissue was highly vascularized (Figure 3f). Multiple predominantly capillary-type blood vessels were identified, and their density was 131.7 ± 37.1 per 1 mm^2^ of the tissue. Signs of active collagen production were noted. Lymphocyte and macrophage infiltration was observed under the gingival epithelium and in isolated foci at the implantation site. No signs of purulent inflammation or necrosis were found.

In the group of Mucoderm in combination with the pCMV-VEGF165 plasmid (‘Neovasculgen’), the submucosal tissue had a larger volume than in the other experimental groups (Figure 2g,h). Collagen fibers were partially disintegrated, and edema was evident in the tissue (Figure 3g). Inclusions of squamous epithelium were identified. The thickness of the submucosa was 3257.8 ± 1093.5 µm. Blood vessels at the implantation site were noticeably different in density and morphology from the other experimental groups—the density of blood vessels was the highest in the experiment, and many of them had a larger caliber (Figure 3h). Many arteries with thick walls were identified; the vessel density was 226.7 ± 45.9 per 1 mm^2^ of tissue. Lymphocyte and macrophage infiltration was diffuse; there were no signs of purulent inflammation or foci of necrosis.

The use of Mucoderm in combination with the pCMV-VEGF165 plasmid (‘Neovasculgen’) potentiated the pro-proliferative effect of the membrane and the pro-vascularization effect of the plasmid (Figure 4). The thickness of the implantation site of the combined membrane with the plasmid was significantly higher than in other groups in the experiment, while blood vessels were formed in greater density and in larger calibers than in other groups. No systemic effects on the vascular system were observed outside the areas of the implantation.

## 3. Discussion

Plasmids carrying the VEGF-165 gene to stimulate angiogenesis are widely used in the therapy of ischemic diseases, both in animal models and clinical practice. Existing plasmids are being improved by adding other angiogenesis-regulating and proliferation genes, such as *FGF2* [21], *HGF* [22], *ANG-1* [23,24], and *SDF-1alpha* [25]. These plasmids are integrated into various scaffolds to accelerate implant integration with vascularization stimulation [26,27]. Primarily, modified scaffolds with the addition of plasmids carrying the *VEGF* gene find application in burn therapy for dermal reconstruction [23,28,29]. The integration of *VEGF* gene plasmids into scaffolds has only been applied in pilot studies for bone tissue reconstruction [30,31]. The use of the VEGF plasmid in the form of injections of the gene therapeutic agent pCMV-VEGF165 plasmid (‘Neovasculgen’) for gingiva soft tissue volume restoration was carried out for the first time in this study.

Plasmids containing the vascular endothelial growth factor (*VEGF*) gene represent a well-established and commonly utilized strategy for promoting vascularization in both in vitro and in vivo settings. Previous studies have demonstrated the impact of these plasmids on the development of capillary-like structures [25] as well as on angiogenesis from endothelial progenitor cells [32]. Furthermore, they are extensively employed in the management of ischemic injuries [33,34], skin wounds [23,29], and stenotic conditions [35].

The pCMV-VEGF165 plasmid (‘Neovasculgen’) was approved for clinical trials to treat lower limb ischemia of atherosclerotic origin, including chronic critical lower limb ischemia (ClinicalTrials.gov identifier: NCT03068585). The first clinical safety and efficacy study involved observing a patient with non-healing reconstructed lower jaw wounds [36]. In clinical studies on 65 patients with lower limb critical ischemia and diabetes mellitus, the addition of two courses of intramuscular injections of the pCMV-VEGF165 plasmid at a dose of 2.4 mg per course showed a positive effect for at least one year of observation [37]. In other studies involving 121 patients with stage IIB-III lower limb chronic ischemia and 45 patients with chronic lower limb ischemia stages II and III, good drug tolerability, absence of side effects, and sustained prolonged positive effects for 3 and 5 years of observation were established, respectively [38,39]. In a clinical study with 62 patients suffering from pyonecrotic complications of Wagner grade III-IV diabetic foot syndrome, a positive effect was demonstrated by adding the pCMV-VEGF165 plasmid (‘Neovasculgen’) to complex therapy [39]. In a clinical case involving a patient suffering from thromboangiitis obliterans (Buerger’s disease) for a long time, ‘Neovasculgen’ as part of comprehensive conservative therapy showed a successful therapeutic outcome [40].

In this study, Mucoderm was used as a scaffold. It is designed for peri-implant soft tissue augmentation procedures. On the one hand, this material based on a decellularized porcine dermis allows for avoiding traumatic autograft harvesting and accelerates soft tissue restoration. It has shown effectiveness in soft tissue restoration during vestibuloplasty [41]. On the other hand, Mucoderm does not promote vascular formation at the site of implantation, potentially leading to challenges with wound healing in the oral cavity [42] and root coverage procedures in instances of gingiva recession [43]. In this study, both the Mucoderm material itself and its combination with the plasmid containing the *VEGF* gene (pCMV-VEGF165 plasmid (‘Neovasculgen’)) were investigated for their efficacy in restoring soft gingiva tissue volume through an integrated approach. The scaffold’s porous structure and structural support enhanced the formation of a microenvironment conducive to vessel growth, while the pCMV-VEGF165 plasmid promoted vessel sprouting at the implantation site and vascularization of newly generated tissue. The administration of the pCMV-VEGF165 plasmid (‘Neovasculgen’) via gingiva injection resulted in a significant local concentration of VEGF in the tissues surrounding the implanted material, facilitating the production of a stimulating plasmid product within host cells. The scaffold and the drug did not affect the metabolic and proliferative activity of human gingiva MSCs; cells adhered well to Mucoderm and formed a dense monolayer on the material’s surface within 7 days (Figure 1).

Tests of metabolic activity, viability of cell proliferation on scaffolds, and in vitro interaction with the drug were performed on mesenchymal stromal cells (MSCs) isolated from the mucous membrane of the free human gingiva according to proven protocols [44]. MSC derived from gingiva and their secretion products are profoundly characterized [45,46,47] and have been widely used in similar in vitro tests of human-related materials [9,48] and tissue engineering [49].

It is noteworthy that the injection of the pure pCMV-VEGF165 plasmid (0.12 mg as per the manufacturer’s protocol) into the soft tissues of the gingiva in rabbits led to the development of keratinizing epithelial structures. In contrast, this phenomenon was not observed when the pCMV-VEGF165 plasmid was used in conjunction with Mucoderm. This suggests that high doses of the drug may have minor toxic effects. It is advisable to consider administering VEGF165 plasmid in multiple smaller injections to reduce the localized concentration of the substance in the gingiva area. This approach may help mitigate potential adverse effects associated with high single-dose concentrations. Despite this, the combination of Mucoderm and the pCMV-VEGF165 plasmid (‘Neovasculgen’) showed excellent results in increasing gingiva volume (submucosa 3257.8 ± 1093.5 µm) and vascularization (226.7 ± 45.9 vessels per 1 mm^2^) of tissues at the site of implantation and injection in comparison with groups of pure matrices (2041.6 ± 496.8 µm; 131.7 ± 37.1 per 1 mm^2^) and the drug (1366.5 ± 420.6 µm; 145 ± 37.82 per 1 mm^2^) (Figure 4). In our experiments with Mucoderm scaffolds and the pCMV-VEGF165 plasmid, we noticed a proliferative tissue response. Our results indicate that *VEGF* gene-containing plasmid and collagen scaffolds can cause stimulation fibroblast and keratinocyte proliferation at the outer edge of the inflammatory area, thereby aiding in the establishment of a new tissue barrier. The occurrence of this response appears to be advantageous, as VEGF appears to enhance endothelial cell migration and capillary formation by promoting the fusion of extracellular spaces between neighboring endothelial cells, emphasizing its critical role as a key regulator of the proliferative phase in inflammation-induced tissue regeneration. We reported that the effects of the plasmid were primarily local, which is important for the safety of the gene therapy. While there is theoretical and experimental evidence suggesting that *VEGF* plasmids could contribute to tumor growth under certain conditions, clinical studies have not demonstrated a significant increase in tumor incidences with the therapeutic use of VEGF plasmids [50]. The synergetic effect of Mucoderm and the pCMV-VEGF165 plasmid demonstrated in this study on the restoration of the volume of soft gingival tissues may be related to the fact that the scaffold provides a suitable three-dimensional structure for cell infiltration and new tissue formation, creating a framework for granulation tissue, while the pCMV-VEGF165 plasmid enhances vascularization, which is critical for the survival and integration of the newly formed tissue. The addition of the pCMV-VEGF165 plasmid enhances signaling in the area of the defect that is ready for vascularization. Similar effects of regeneration stimulation through the activation of epithelialization, collagen synthesis, proliferation, and angiogenesis with immunomodulatory action have previously been demonstrated with the use of collagen materials and plasmid DNA encoding VEGF-165 in wound healing [51,52], including diabetic chronic wounds [53] and stimulation of repair in extensive bone tissue defects [30,54]. Further analysis of the expression of angiogenesis-related proteins and the dynamics of changes in the ratio of different cell types during regeneration compared to the gold standard of surgical treatment, the use of a connective tissue graft from the palate, will fully elucidate the mechanism of the demonstrated synergetic effect of Mucoderm and the pCMV-VEGF165 plasmid on the restoration of the volume of soft gingival tissues.

In our study, we demonstrated a joint positive effect on gingival tissue augmentation and stimulation of vascularization with simultaneous use of Mucoderm and the pCMV-VEGF165 plasmid (Figure 5).

The findings of this study represent a significant step towards the clinical application of gene-therapeutic and scaffold-based approaches for gingival tissue augmentation. However, given the anatomical, histological, and physiological differences between rabbit and human gingival tissues, additional steps are necessary to translate these findings to human clinical trials. Rabbits exhibit faster wound healing and a different immune response compared to humans, which may influence the outcomes observed in this study [55]. To address these differences, further preclinical studies should be conducted using gingival tissue models, such as ex vivo human gingiva, or by conducting gingival implantations in minipigs. Additionally, the pharmacokinetics and dosage optimization of the pCMV-VEGF165 plasmid for human applications need thorough investigation to ensure safety and efficacy. These studies should be complemented by evaluating the long-term effects and potential immunogenicity of the combined treatment. Once these aspects are sufficiently addressed, the translation to human clinical trials can be more confidently pursued.

This study provides valuable insights into the potential therapeutic applications of the pCMV-VEGF165 plasmid in combination with Mucoderm for gingiva soft tissue volume restoration. Furthermore, these findings contribute to the growing body of evidence supporting the use of gene therapeutic agents in tissue engineering and regenerative medicine.

## 4. Materials and Methods

### 4.1. Mucoderm and ‘Neovasculgen’ Materials

Mucoderm is a volumetric soft tissue graft made from porcine skin (Botiss, Berlin, Germany). For both in vitro and in vivo experiments, we used pieces that were 0.5 cm^2^ (0.7 × 0.7 cm). In the manufacturer’s packaging, Mucoderm is represented by a dry sterile matrix. For in vitro tests, it was cut and soaked in a growth medium to obtain extracts, and before starting in vivo experiments on a rabbit model, it was pre-soaked in a saline solution. The pCMV-VEGF165 plasmid (‘Neovasculgen’) (Haematology Research Centre, Moscow, Russia) is a lyophilized plasmid carrying the human *VEGF 165* gene for producing VEGF (vascular endothelial growth factor) and stimulating the growth of vessels. ‘Neovasculgen’ in the manufacturer’s packaging is represented by a dry lyophilized plasmid powder. To prepare solutions for in vitro and in vivo tests, the lyophilized plasmid was dissolved in sterile solutions of growth medium and water for injection in accordance with the manufacturer’s recommendations. For in vitro experiments, we used concentrations of plasmid ranging from 0.001 mg/mL to 0.12 mg/mL, and for in vivo experiments, we used 500 µL of solution, which contained 0.12 mg of plasmid (according to the manufacturer’s protocol).

### 4.2. Cell Culturing

The primary culture of human gingival mesenchymal stem cells obtained from the Biobank of Sechenov University (Moscow, Russia) was used for the biocompatibility study. Obtained MSCs were cultured in Dulbecco’s Modified Eagle’s Medium (DMEM)/F12 (1:1, Biolot, St. Petersburg, Russia) supplemented with 10% fetal calf serum (HyClone, Wilmington, NC, USA), insulin–transferrin–sodium selenite (1:100, Biolot, Saint-Petersburg, Russia), bFGF (20 ng/mL, ProSpec, Ness-Ziona, Israel), heparin 5000 IU/mL (Belmedpreparaty, Minsk, Belarus), and gentamicin (50 g/mL, Paneco, Moscow, Russia) at 37 °C and 5% CO_2_. The cells were maintained under standard conditions (37 °C, 95% humidity, 5% CO_2_) and routinely examined using a phase-contrast microscope Axio Vert A1 (Carl Zeiss, München, Germany), with the medium replaced every 3 days. 

### 4.3. Biocompatibility

The biocompatibility of Mucoderm and ‘Neovasculgen’ was evaluated through extract cytotoxicity tests using AlamarBlue and PicoGreen assays. Additionally, contact culturing on these materials was conducted to enable subsequent Live/Dead assay analyses.

### 4.4. Extract Cytotoxicity

To assess the biocompatibility of the materials, we conducted a series of tests using extracts and mesenchymal stem cells (MSCs). The cell culturing process followed the protocol outlined in the “Cell Culturing” section. MSCs were seeded in triplicate in 96-well plates (Corning, Glendale, AZ, USA) at a concentration of 5000 cells per well. The extracts were prepared according to the method described in [56] by incubating 6 cm^2^ samples of Mucoderm in 1 mL of growth medium at 37 °C for 24 h. ‘Neovasculgen’ was prepared at a maximal concentration of 0.12 mg/mL following the manufacturer’s instructions using the growth medium. Sodium dodecyl sulphate (SDS) dilutions served as a positive control, and the maximal concentration was 1.5 mg/mL. Serial dilutions of Mucoderm extract, ‘Neovasculgen’, and sodium dodecyl sulfate (SDS) ranging from 100% to 0.78%, were prepared and added to the MSCs seeded in the 96-well plates (Corning, Glendale, AZ, USA); full growth medium was used as a negative control. The cells were then incubated with the extracts and other additives (SDS, ‘Neovasculgen’) for 24 h at 37 °C in 5% CO_2_.

The AlamarBlue cell viability reagent (Invitrogen, Waltham, MA, USA) was employed to assess the metabolic activity of the cells. After removing the growth medium from the wells, 100 µL of AlamarBlue reagent was added according to the manufacturer’s instructions and incubated for 2 h at 37 °C, 5% CO_2_ in the dark. Subsequently, the plates were read on a Victor Nivo spectrofluorometer (PerkinElmer, Waltham, MA, USA) at an excitation wavelength of 580/20 nm and an emission wavelength of 625/30 nm. Cell proliferation and double-stranded DNA amount were assessed using the Quant-iT PicoGreen dsDNA Assay Kit (Thermo Fisher Scientific, Waltham, MA, USA).

For the PicoGreen cytotoxicity test, the AlamarBlue cell viability reagent was removed from the wells, and the cells were washed three times with a PBS solution. Next, 100 µL of distilled H_2_O was added to each well, and the plates were frozen at −20 °C three times. Subsequently, PicoGreen reagent was added to each well and incubated at room temperature in the dark for 15 min. The plates were then read on a Victor Nivo spectrofluorometer (PerkinElmer, Waltham, MA, USA) at an excitation wavelength of 480/30 nm and an emission wavelength of 530/30 nm.

### 4.5. Contact Culturing (Live/Dead Assay)

Sterile samples of Mucoderm were seeded with 100,000 cells per 0.5 cm^2^ (0.7 × 0.7 cm). Samples were cultured for 3 and 7 days under standard conditions (37 °C, 5% CO_2_), and then cell viability was assessed using the Live/Dead assay (0.5 mg/mL, Calcein-AM, Sigma-Aldrich, Burlington, MA, USA; 1.5 μM, propidium iodide, Thermo Fisher Scientific, Waltham, MA, USA). Live cells were stained with Calcein AM (green) (Thermo Fisher Scientific), dead cells were stained with propidium iodide (red), and nuclei were additionally stained with Hoechst (blue). After staining, samples were washed with a serum-free DMEM/F-12 medium and then placed in confocal dishes in a full growth medium. Visualization was performed on a laser scanning confocal microscope LSM 880 (Carl Zeiss, Oberkochen, Germany) using excitation lasers at 405 nm (Hoechst 33258, Thermo Fisher Scientific), 488 nm (Calcein AM, Thermo Fisher Scientific), and 594 nm (propidium iodide).

### 4.6. Scanning Electron Microscopy

The Mucoderm matrices with and without cells were prepared for scanning electron microscopy (SEM) by first fixing them with a 3% glutaraldehyde solution in phosphate-buffered saline (PBS) overnight at +4 °C. Subsequently, the samples were washed three times with PBS for 5 min each and then fixed in a 1% osmium tetroxide (OsO4) solution in PBS for 40 min at room temperature. Following this, both cell-seeded and cell-free Mucoderm samples underwent three additional washes with PBS and were then dehydrated using a series of ethanol washes (twice with 50% ethanol for 5 min each, twice with 70% ethanol for 10 min each, once with 80% ethanol for 5 min, and twice with 95% ethanol for 5 min each) and acetone (twice for 5 min each). The samples were stored in 70% ethanol. Matrices were then dried at the critical point, coated with gold under vacuum conditions, and examined using a CamScan-S2 scanning electron microscope (Cambridge Instruments, Cambridge, UK) to study the replicas.

### 4.7. Animal Experiments

In this experimental study, female rabbits of the “Gray Giant” breed weighing 3000–3200 g were used, with 12 individuals divided into four groups of three individuals each. The animals were categorized into the following groups: Group 1 (Mucoderm), Group 2 (‘Neovasculgen’), Group 3 (Mucoderm + ‘Neovasculgen’), and Group 4 (Intact gingiva control). The effectiveness of gingival soft tissue restoration was compared after closed gingival mucosa augmentation with implantation of the material and/or injection of the drug under the flap in rabbits of the breed. Surgical intervention was carried out targeting the upper jaw on both sides of the vestibular surface of the gingival mucosa. A linear incision was made, followed by apical movement and fixation of the superficial flap with sutures. The examined material was placed on the periosteum and secured with double nodular sutures, depending on the group assignment, with the exception of Group 2, where only ‘Neovasculgen’ injection was administered under the mucosa. ‘Neovasculgen’ was injected intraoperatively into the implantation area, with no more than 50 µL per injection site. All manipulations were performed under conditions of medication analgesia using a solution of tiletamine and zolazepam (ZOLETIL 100) and a solution of medetomidine (Meditin). Before surgery, animals were anesthetized intramuscularly with combined anesthesia ZOLETIL 100 (VIRBAC, Carros, France) at a dosage of 15 mg/kg and Meditin (Apicenna, Moscow, Russia) at a dosage of 0.5 mg/kg. On the 14th postoperative day, the rabbits were euthanized by the injection of a solution of ZOLETIL 100 (VIRBAC, France) at a dosage of 60 mg/kg. The sites of implantation were dissected together with 2–3 mm of surrounding tissues. In order to investigate systemic effects of the implants, a kidney, a liver and a complex of heart and lungs were collected from all animals. The tissues and organs were fixed in neutral buffered formalin for 24–48 h.

### 4.8. Histological Analysis

Four μm thick sections of the formalin-fixed-paraffin-embedded tissue samples were stained with hematoxylin and eosin (H&E) and with Masson trichrome for the detection of collagen fibers. A LEICA DM4000 B LED microscope, equipped with a LEICA DFC7000 T digital camera running under the LAS V4.8 software (Leica Microsystems, Wetzlar, Germany), was used for the examination and imaging of the samples. 

For immunohistochemical analysis, four μm thick sections of the formalin-fixed-paraffin-embedded tissue samples were deparaffinized, incubated in 3% hydrogen peroxide (H_2_O_2_) for 10 min, underwent heat-induced epitope retrieval (pH 6.0 sodium citrate buffer, 30 min in 80 °C water bath), additionally blocked with Background Block (Cell Marque, Rocklin, CA, USA) and incubated with mouse monoclonal primary antibodies against α-smooth muscle actin (α-SMA) (A2547, Merck Millipore, Burlington, MA, USA, diluted 1:400), and detected by HRP-conjugated secondary goat antibodies (G-21040, Invitrogen, Carlsbad, CA, USA, diluted 1:1000) and diaminobenzidine (DAB) with hematoxylin counterstaining.

Blood vessel density was evaluated at ×200 magnification in 10 representative fields of view. The results of the blood vessel density analysis were counted as average number per 1 mm^2^ for each sample. The thickness of the gingival submucosa was evaluated as an average of 5 measurements at a distance of 400 μm from each other.

The statistical analysis of the experimental data was performed with a standard program package, GraphPad Prism version 8.00 for Windows (GraphPad Software, Inc., San Diego, CA, USA). The intergroup differences were analyzed by one-way ANOVA followed by Tukey’s multiple comparison test. The statistical analysis results were presented as column graphs of the mean values and standard deviations (SDs). *p*-values equal to or less than 0.05 were considered statistically significant.

## 5. Conclusions

In this study, we achieved an increase in the volume of soft gingival tissues and enhancement of vascularization through the implantation of a xenogeneic matrix made from porcine skin Mucoderm and injections of the pCMV-VEGF165 plasmid (‘Neovasculgen’) in rabbits. The plasmid and matrix did not exhibit toxic effects in the in vitro studies and were fully biocompatible with the gingival mesenchymal stem cell culture based on viability and metabolic activity tests. The use of Mucoderm in combination with the pCMV-VEGF165 plasmid (0.12 mg; manufacturers protocol) potentiated the pro-proliferative effect of the membrane and the pro-vascularization effect of the plasmid. The thickness of the soft tissues at the implantation site was significantly greater when Mucoderm and the pCMV-VEGF165 plasmid were combined. The combined application of the Mucoderm and pCMV-VEGF165 plasmid resulted in a significant increase in both the density of blood vessels and their average diameter. These findings suggest promising prospects for the clinical application of this innovative approach in regenerative medicine and dentistry.

## Figures and Tables

**Figure 1 ijms-25-10013-f001:**
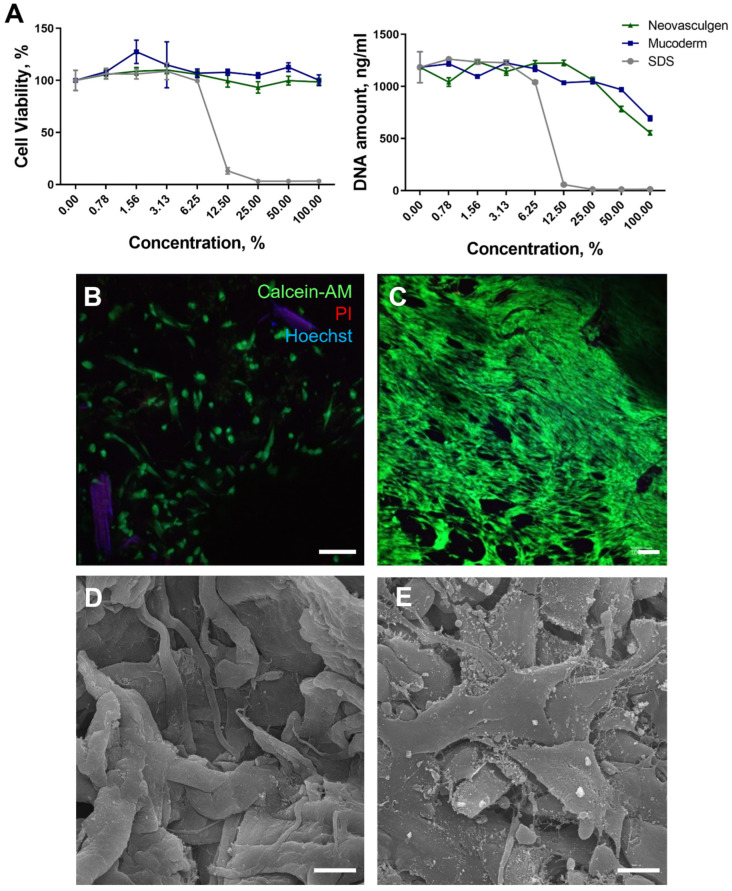
Evaluation of the MSC viability on the Mucoderm surface and pCMV-VEGF165 plasmid (‘Neovasculgen’) extracts. (**A**): Relative cell viability curves with AlamarBlue test and PicoGreen tests for samples’ extracts: Mucoderm and pCMV-VEGF165 plasmid (‘Neovasculgen’). Sodium dodecyl sulphate (SDS) serial dilutions (light grey line) were used as a positive control. (**B**,**C**): Live/Dead assay with additional nuclei staining. Living cells were stained with Calcein-AM (green), dead cells were stained with Propidium Iodide (PI) (red), and nuclei were stained with Hoechst (blue) at 3 days of cultivation (**B**) and 7 days of cultivation (**C**). The scale bar is 100 µm. (**D**,**E**): SEM photos of Mucoderm material without (**D**) and with MSCs (**E**) on the surface after 3 days of cultivation. The scale bar is 10 µm.

**Figure 2 ijms-25-10013-f002:**
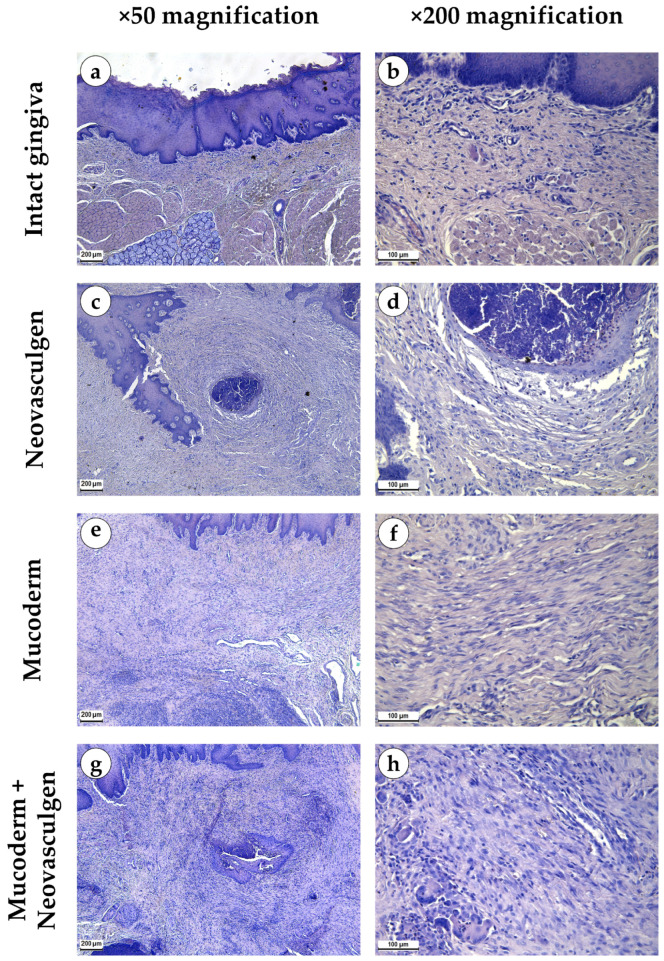
Morphological analysis of intact gingival tissues (**a**,**b**), areas of injection of Neovasculgen (**c**,**d**), implantation of Mucoderm (**e**,**f**) and areas of implantation of Mucodern followed by Neovasculgen injection (**g**,**h**), hematoxylin and eosin, magnifications ×50 and ×200.

**Figure 3 ijms-25-10013-f003:**
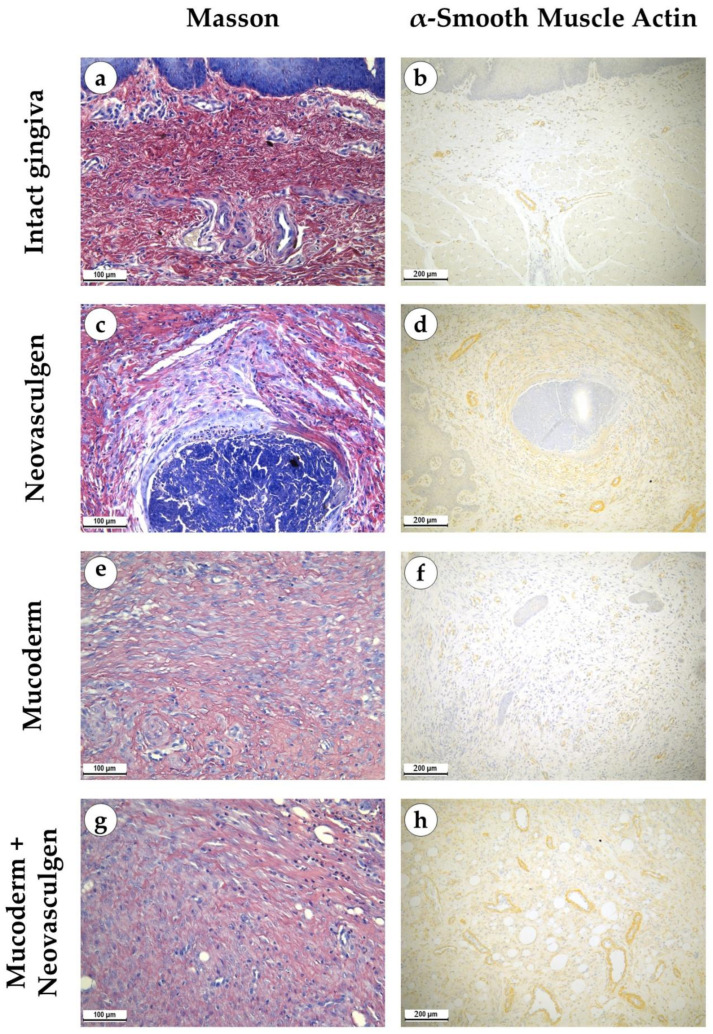
Morphological analysis of intact gingival tissues (**a**,**b**), areas of injection of Neovasculgen (**c**,**d**), implantation of Mucoderm (**e**,**f**) and areas of implantation of Mucodern followed by Neovasculgen injection (**g**,**h**), Masson trichrome staining (**a**,**c**,**e**,**g**), immunohistochemical reaction with antibodies against α-smooth muscle actin (**b**,**d**,**f**,**h**), magnifications ×200 and ×100.

**Figure 4 ijms-25-10013-f004:**
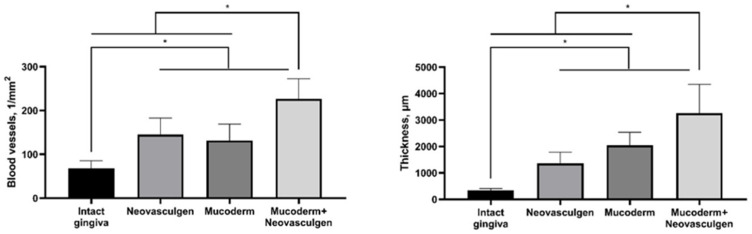
Statistical analysis of blood vessel density and submucosal thickness in rabbit gingival tissue. One-way ANOVA, mean values ± SD. * *p* ≤ 0.05.

**Figure 5 ijms-25-10013-f005:**
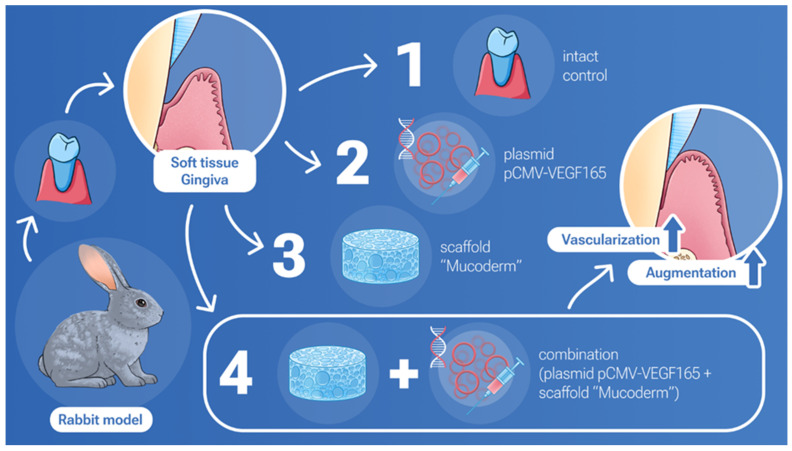
Integrated results of soft tissue augmentation: evaluating the impact of Mucoderm and the VEGF plasmid alone and in combination.

## Data Availability

Data is contained within the article.

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
