# Peer review of "Gene Therapeutic Drug pCMV-VEGF165 Plasmid (‘Neovasculgen’) Promotes Gingiva Soft Tissue Augmentation in Rabbits"

_ijms, 2024, doi:10.3390/ijms251810013_

Round 1
Reviewer 1 Report
Comments and Suggestions for Authors
This article presents a novel strategy for enhancing the volume and vascularization of rabbit soft gingival tissue and confirms the biocompatibility and therapeutic potential of this “combination of Mucoderm and pCMV-VEGF165 17 plasmid” through a series of in vitro and in vivo experiments. After addressing the following issues, I recommend its publication.
1. The summary part is quite simple, please expand it appropriately.
2. Are the Mucoderm and pCMV-VEGF165 plasmid the ready-made products that purchased directly? If not, please give the specific preparation process.
3. The results presented in Fig 1A to 1D do not correspond to the experimental results described in the article. Please check this issue carefully.
4. The expression of Fig 1A to 1D is extremely unreasonable. Please re-sort the relevant results and insert clearer and more explicit data charts.
5. Why do Fig 2, Fig 3 and Fig 4 not correspond to the relevant content of the article? Please add these annotations to the text.
6. There are a number of formatting errors in the article, please check them carefully and correct them.
7. The discussion section of the article is good, but a little boring, adding relevant data pictures is recommended to better present these discussions to readers.
8. There are some relevant reviews that authors are advised to cite references to enhance the statement of the manuscript.
https://doi.org/10.1007/s12274-022-5129-1ï¼›https://pubs.rsc.org/en/content/articlehtml/2024/cs/d3cs00923h
Comments on the Quality of English Language
Minor editing of English language required.
Author Response
- The summary part is quite simple, please expand it appropriately.
Responce 1:
Thank you very much for the important recommendation. The summary has been expanded and supplemented. - Are the Mucoderm and pCMV-VEGF165 plasmid the ready-made products that purchased directly? If not, please give the specific preparation process.
Responce 2:
Thank you for your note. We added the following information about Mucoderm and Neovasculgen preparation process to the Methods:
«In the manufacturer's packaging, Mucoderm is represented by a dry sterile matrix, for in vitro tests it was cut and soaked in a growth medium to obtain extracts, and before starting in vivo experiments on a rabbit model, it was pre-soaked in saline solution.»
«'Neovasculgen' in the manufacturer's packaging is represented by dry lyophilized plasmid powder. To prepare solutions for in vitro and in vivo tests, the lyophilized plasmid was dissolved in sterile solutions of growth medium and water for injection in accordance with the manufacturer's recommendations.» - The results presented in Fig 1A to 1D do not correspond to the experimental results described in the article. Please check this issue carefully.
Responce 3:
Thank you for your valuable feedback on the discrepancies in Figures 1A to 1D. We have reviewed the data and text, making the necessary corrections to ensure accuracy in Results section, “Biocompatibility of Mucoderm and pCMV-VEGF165 plasmid”. - The expression of Fig 1Ato 1Dis extremely unreasonable. Please re-sort the relevant results and insert clearer and more explicit data charts.
Responce 4:
Thank you for your valuable feedback. We have taken your suggestion into account and have added some information to the Methods and reorganized the graphs to present the data for each test separately, which we believe enhances clarity for readers. In our study, we assessed cell viability using the AlamarBlue reagent and also evaluated DNA quantity with the PicoGreen assay. The results from both assays indicate that neither Mucoderm extracts nor the pCMV-VEGF165 plasmid had any adverse effects on cell viability. - Why do Fig 2, Fig 3and Fig 4 not correspond to the relevant content of the article? Please add these annotations to the text.
Responce 5:
We added references to each subfigure to relevant places in the text. - There are a number of formatting errors in the article, please check them carefully and correct them.
Responce 6:
Thank you very much for your valuable feedback. We have carefully reviewed our manuscript and made the necessary corrections to address the errors identified. We hope these revisions enhance the clarity and readability of our work for our esteemed readers. - The discussion section of the article is good, but a little boring, adding relevant data pictures is recommended to better present these discussions to readers.
Responce 7:
We added a scheme to better visualize the main results of the study (fig. 5). The discussion also has been expanded and supplemented with information on the synergy of combined application of VEGF plasmids and collagen scaffolds, as well as the prospects for implementing this aproach in dental clinical practice. - There are some relevant reviews that authors are advised to cite references to enhance the statement of the manuscript.
https://doi.org/10.1007/s12274-022-5129-1ï¼›https://pubs.rsc.org/en/content/articlehtml/2024/cs/d3cs00923h
Responce 8:
Thank you, to enchance the statement for the approaches used to stimulate angiogenesis and regeneration, information, including that from the recommended reviews, has been added to the Introduction.

Reviewer 2 Report
Comments and Suggestions for Authors
This paper discusses a study on the use of a gene therapeutic drug pCMV-VEGF165 plasmid for promoting gingiva soft tissue augmentation in rabbits, in conjunction with a xenogeneic scaffold derived from porcine skin Mucoderm.
This study investigated the potential of combining Mucoderm derived from porcine skin with injections of pCMV-VEGF165 plasmid to enhance soft gingival tissue volume and vascularization in rabbits. In vitro studies demonstrated the biocompatibility of the matrix and plasmid with gingival mesenchymal stem cells, showing no toxic effects. The in vivo experiments showed that the combination of Mucoderm and pCMV-VEGF165 plasmid significantly increased the thickness of soft tissues at the implantation site and augmented the density and diameter of blood vessels. The combined application resulted in a synergistic effect, potentiating both the pro-proliferative effect of the membrane and the pro-vascularization effect of the plasmid. The authors conclude that this approach shows promising potential for clinical applications in regenerative medicine and dentistry, particularly for soft tissue augmentation and improved vascularization in gingival tissues. Here are some clarifications that can be added to make the statements clearer:
1. What is the mechanism behind the synergistic effect of Mucoderm and the VEGF plasmid? One hypothesis could be that the Mucoderm scaffold provides a suitable three-dimensional structure for cell infiltration and new tissue formation, while the VEGF plasmid enhances vascularization, which is critical for the survival and integration of the newly formed tissue. Please add discussion and address further studies or techniques needed.
2. Were any systemic effects of the VEGF plasmid observed or measured in the rabbits? How was the potential for off-target angiogenesis addressed?
3. For the in-vivo experiment, is there long-term follow-ups to assess the stability of the augmented tissue and vascularization? While the study demonstrates impressive short-term results, the long-term stability of the augmented tissue is crucial for clinical applications. Comparative studies with current gold standard techniques would provide more context into the long-term efficacy of the proposed approach.
4. VEGF is a potent angiogenic factor that plays a crucial role in tumor growth and metastasis. Prolonged or uncontrolled VEGF expression could potentially promote the growth of pre-existing tumors or induce the formation of vascular tumors like hemangiomas. The safety concern of the use of VEGF-expressing plasmids needs to be carefully examed.
5. Given the differences between rabbit and human gingival tissue, it would be of interest to understand how far are the findings towards human application. What additional steps would be needed to translate these findings to human clinical trials?
Comments on the Quality of English Language
Good Quality of Language
Author Response
- What is the mechanism behind the synergistic effect of Mucoderm and the VEGF plasmid? One hypothesis could be that the Mucoderm scaffold provides a suitable three-dimensional structure for cell infiltration and new tissue formation, while the VEGF plasmid enhances vascularization, which is critical for the survival and integration of the newly formed tissue. Please add discussion and address further studies or techniques needed.
Responce 1:
Thank you for the valuable recommendations. The Discussion has been expanded and supplemented with information on the mechanisms of the synergististic effect of combined application of VEGF plasmids and collagen scaffolds. The prospects for further study and practical implementation of the results obtained have also been added to the Discussion section. - Were any systemic effects of the VEGF plasmid observed or measured in the rabbits? How was the potential for off-target angiogenesis addressed?
Responce 2:
We added the following to the Methods:
“On the 14th postoperative day, the rabbits were euthanized by the injection of a solution of ZOLETIL 100 (VIRBAC, France) at a dosage of 60 mg/kg. The sites of implantation were dissected together with 2–3 mm of surrounding tissues. In order to investigate systemic effects of the implants, a kidney, a liver and a complex of heart and lungs were collected from all animals. The tissues and organs were fixed in neutral buffered formalin for 24 – 48 hours.”
We added the following to the Results section:
“No systemic effects on the vascular system were observed outside the areas of the implantation”. - For the in-vivo experiment, is there long-term follow-ups to assess the stability of the augmented tissue and vascularization? While the study demonstrates impressive short-term results, the long-term stability of the augmented tissue is crucial for clinical applications. Comparative studies with current gold standard techniques would provide more context into the long-term efficacy of the proposed approach.
Responce 3:
The aim of the present study was to investigate whether the combination of a matrix with a plasmid can potentiate their pro-regenerative properties. We agree that further pre-clinical studies, including long-term assessments, are necessary to move this research closer to clinical translation. Additionally, we believe that minipigs would be the most appropriate model for the next phase of studies. The Discussion has been expanded with a section on the prospects for further research and the translation of the obtained results in dental clinical practice. - VEGF is a potent angiogenic factor that plays a crucial role in tumor growth and metastasis. Prolonged or uncontrolled VEGF expression could potentially promote the growth of pre-existing tumors or induce the formation of vascular tumors like hemangiomas. The safety concern of the use of VEGF-expressing plasmids needs to be carefully examed.
Responce 4:
We added the following to the Discussion:
“We reported that the effects of the plasmid were primarily local which is important argument for the safety of the gene therapy. While there is theoretical and experimental evidence suggesting that VEGF plasmids could contribute to tumor growth under certain conditions, clinical studies have not demonstrated a significant increase in tumor incidence with the therapeutic use of VEGF plasmids (Giacca, M.; Zacchigna, S. VEGF gene therapy: therapeutic angiogenesis in the clinic and beyond. Gene Ther 2012, 19, 622-629, doi:10.1038/gt.2012.17.).” - Given the differences between rabbit and human gingival tissue, it would be of interest to understand how far are the findings towards human application. What additional steps would be needed to translate these findings to human clinical trials?
Responce 5:
We added the following paragraph to the Discussion:
“The findings of this study represent a significant step towards the clinical application of gene-therapeutic and scaffold-based approaches for gingival tissue augmentation. However, given the anatomical, histological and physiological differences between rabbit and human gingival tissues, additional steps are necessary to translate these findings to human clinical trials. Rabbits exhibit faster wound healing and a different immune response compared to humans, which may influence the outcomes observed in this study (Naeini, A.T.; Oryan, A.; Dehghani, S.; Nikahval, B. Experimental cutaneous wound healing in rabbits: using continuous microamperage low-voltage electrical stimulation. Comparative Clinical Pathology 2008, 17, 203-210). To address these differences, further preclinical studies should be conducted using gingival tissue models, such as ex vivo human gingiva or conducting gingival implantations in minipigs. Additionally, the pharmacokinetics and dosage optimization of the pCMV-VEGF165 plasmid for human applications need thorough investigation to ensure safety and efficacy. These studies should be complemented by evaluating the long-term effects and potential immunogenicity of the combined treatment. Once these aspects are sufficiently addressed, the translation to human clinical trials can be more confidently pursued.”

Reviewer 3 Report
Comments and Suggestions for Authors
In the presented manuscript interesting research of employing gene therapeutic drug pCMV-VEGF165 plasmid in promoting gingiva soft tissue augmentation is presented. It is an in vitro and in vivo study, on gingival mesenchimal stem cells and rabbit as animal model, and biocompatibility, histological and morfometric analysis were carried out.
The Introduction provides all important background information while appropriately identifying relevant references.
The research design is well-thought-out, comprehensive, and follows established principles for in vitro and in vivo research.
The Disscusion provides adequate mentioning of other studies results, comparison to the obtained results and methods used.
References are adequate and up-to-date.
Overall, research is interesting, comprehensive and original. There is a few points to be resolved, so the paper gain on its quality.
In Material and Methods :
For in vitro - why there is no group Mucoderm + ‘Neovasculgen’?
For in vivo - is healthy gingival tissue flap made, material placed and then sutured? There was no any wound formation previously?
How did material was applied in Mucoderm + ‘Neovasculgen’ group?
In Results:
For Figure 1 it would be better for the readers to put legend next to the graph.
In Conclusion:
There is no need of repeating numerical values for gained results.
Author Response
In Material and Methods :
1. For in vitro - why there is no group Mucoderm + ‘Neovasculgen’?
Responce 1:
Thank you for your question. Our decision to evaluate the matrix and plasmid in vitro separately was made to ensure a comprehensive understanding of the safety profile of each component. In our in vivo studies, we indeed utilized Mucoderm in conjunction with Neovasculgen to augment the gingival mucosa and stimulate angiogenesis. The synergistic effects of these materials may only be fully realized during the implantation and injection phases in animal models.
- For in vivo - is healthy gingival tissue flap made, material placed and then sutured? There was no any wound formation previously?
Responce 2:
In the in vivo experiments, the mucous membrane of the gingiva was intact and unchanged prior to the surgical intervention. An incision was made, and a mucoperiosteal flap was reflected. In the experimental groups with collagen scaffolds, the Mucoderm was fixed in the recipient's bed under the mucous flap, and the wound was closed tightly with simple interrupted sutures. - How did material was applied in Mucoderm + ‘Neovasculgen’ group?
Responce 3:
Thank you for your valuable question. To improve readers' understanding of the method, we have added information about ‘Neovasculgen’ plasmid injections to the Methods.
In accordance with the procedure described above and in the Methods section, an incision was made on the intact mucous membrane of the rabbits' gums, and a mucoperiosteal flap was reflected. Mucoderm was fixed in the recipient bed under the mucous flap, and the wound was tightly closed with simple interrupted sutures. ‘Neovasculgen’ 500 µl water solution, which contained 0.12 mg of plasmid (according to the manufacturer’s protocol), was injected intraoperatively into the implantation area, with no more than 50 µl per injection site.
In Results:
4. For Figure 1 it would be better for the readers to put legend next to the graph.
Responce 4:
Thank you for your valuable feedback. We reorganized the graphs and added legend, which we believe enhances clarity for readers.
In Conclusion:
5. There is no need of repeating numerical values for gained results.
Responce 5:
Thank you for your insightful comments regarding our manuscript. We appreciate your feedback and have taken your suggestion into account. As requested, we have removed the numerical values from the Conclusion section to enhance focus.
